# Trazodone, dibenzoylmethane and tauroursodeoxycholic acid do not prevent motor dysfunction and neurodegeneration in Marinesco-Sjögren syndrome mice

Giada Lavigna[1], Anna Grasso[1], Chiara Pasini[1], Valentina Grande[1¤a], Laura Mignogna[1], Elena Restelli[1¤b], Antonio Masone[1], Claudia Fracasso[2], Jacopo Lucchetti[2], Marco Gobbi[2], Roberto Chiesa[1]*

1 Department of Neuroscience, Laboratory of Prion Neurobiology, Istituto di Ricerche Farmacologiche Mario Negri IRCCS, Milan, Italy, 2 Department of Molecular Biochemistry and Pharmacology, Laboratory of Pharmacodynamics and Pharmacokinetics, Istituto di Ricerche Farmacologiche Mario Negri IRCCS, Milan, Italy

¤a Current address: Department of Experimental Neurodegeneration, Center for Biostructural Imaging of Neurodegeneration, University Medical Center Göttingen, Göttingen, Germany
¤b Current address: Human Technopole, Milan, Italy
* roberto.chiesa@marionegri.it

**Data Availability Statement:** The data underlying the results presented in the study are included in the supplementary information.

## Abstract

There is no cure for Marinesco-Sjögren syndrome (MSS), a genetic multisystem disease linked to loss-of-function mutations in the *SIL1* gene, encoding a BiP co-chaperone. Previously, we showed that the PERK kinase inhibitor GSK2606414 delays cerebellar Purkinje cell (PC) degeneration and the onset of ataxia in the woozy mouse model of MSS. However, GSK2606414 is toxic to the pancreas and does not completely rescue the woozy phenotype. The present study tested trazodone and dibenzoylmethane (DBM), which partially inhibit PERK signaling with neuroprotective effects and no pancreatic toxicity. We also tested the chemical chaperone tauroursodeoxycholic acid (TUDCA), which protects MSS patients' cells from stress-induced apoptosis. Mice were chronically treated for five weeks, starting from a presymptomatic stage. Trazodone was given 40 mg/kg daily by intraperitoneal (ip) injection. DBM was given 0.5% in the diet ad libitum. TUDCA was given either 0.4% in the diet, or 500 mg/kg ip every three days. None of the treatments prevented motor dysfunction or PC degeneration in woozy mice, as assessed by beam walking, rotarod test, and calbindin immunohistochemistry. Only trazodone slightly boosted beam walking performance, but this effect was not related to inhibition of PERK signaling. Pharmacokinetic studies excluded that the lack of effect was due to altered drug metabolism in woozy mice. These results indicate that trazodone, DBM and TUDCA, at dosing regimens active in other neurodegenerative disease mouse models, have no disease-modifying effect in a preclinical model of MSS. This underscores the difficulty of translating neuroprotective strategies from other conditions to MSS, highlighting the need for more targeted therapeutic approaches.

**Funding:** "This work was funded by Fondazione Telethon (GGP20071 to RC). Maintenance of the woozy mouse colony was partially supported by the European Union - Next Generation EU - NRRP M6C2 - Investment 2.1 Enhancement and strengthening of biomedical research in the NHS (PNRR-MR1-2022-12375730). The funders had no role in study design, data collection and analysis, decision to publish, or preparation of the manuscript."

**Competing interests:** The authors have declared that no competing interests exist.

## Introduction

Marinesco-Sjögren syndrome (MSS; OMIM 248800) is a rare autosomal recessive disorder with onset in early infancy, causing mainly cerebellar ataxia, progressive myopathy, cataracts and intellectual disability [1]. Approximately 60% of MSS patients carry homozygous or compound heterozygous *SIL1* variants that make the SIL1 protein metabolically unstable, eventually leading to its loss [2–4].

The SIL1 protein is an ATP-exchange factor for the endoplasmic reticulum (ER) chaperone binding immunoglobulin protein (BiP), which plays a central role in protein folding [5]. BiP's ability to bind unfolded proteins and release the folded substrate is tightly regulated by a cycle of ATP binding, hydrolysis, and nucleotide exchange. SIL1 binds to ADP-bound BiP to catalyze the release of ADP, allowing rebinding of ATP. If this nucleotide exchange is defective, as in the case of SIL1 loss, BiP remains bound to its client protein, ultimately leading to accumulation of unfolded proteins, ER stress and activation of the unfolded protein response (UPR) [6,7].

The UPR is a complex signaling pathway whose purpose is to restore ER proteostasis [8]. The kinase PERK is one of the three primary effectors of the UPR, the others being ATF6 and IRE1. Activated PERK phosphorylates the alpha subunit of eukaryotic translation initiation factor 2 (eIF2$\alpha$). This inhibits mRNA translation, reducing global protein synthesis. Some specific mRNA, however, are preferentially translated, including the transcription factor ATF4. If cells are unable to handle the unfolded protein load, protracted ATF4 synthesis induces C/EBP Homologous Protein (CHOP), leading to apoptosis.

Chronic activation of the PERK/eIF2$\alpha$ branch of the UPR is especially detrimental to neurons, which are dependent on new protein synthesis for synaptic maintenance and survival [9]. Thus genetic and pharmacological inhibition of this pathway to restore synaptic protein synthesis has marked neuroprotective effects in mouse models of protein-misfolding diseases [10].

We found that PERK/eIF2$\alpha$ signaling prefigured Purkinje cell (PC) degeneration and ataxia in woozy mice [11], which carry a spontaneous *Sil1* mutation and recapitulate key pathological features of MSS [6,7]. These observations suggested that inhibiting PERK signaling might be beneficial in MSS. Supporting this, presymptomatic treatment of woozy mice with the PERK inhibitor GSK2606414 significantly delayed neurodegeneration and the onset of motor deficits, prolonging the asymptomatic phase of the disease [11]. However, GSK2606414 is toxic to the pancreas and does not provide complete neuroprotection, possibly because full translational recovery due to complete PERK inhibition overloads the inefficient folding machinery of SIL1-deficient cells. It may be necessary to fine-tune PERK signaling and boost protein folding in the ER for better effects [3,12].

The licensed antidepressant trazodone and the licorice derivative dibenzoylmethane (DBM) partially inhibit PERK signaling downstream of eIF2$\alpha$-P, and have neuroprotective effects in prion disease and tauopathy mouse models similar to GSK2606414, but without pancreatic toxicity [13]. By only partially rescuing translation, trazodone and DBM may promote synthesis of pro-survival proteins without overloading SIL1-deficient cells with unfolded proteins.

Tauroursodeoxycholic acid (TUDCA) is an endogenous bile acid with chemical chaperone activity and was reported to attenuate ER stress-induced apoptosis in lymphoblastoid cells derived from MSS patients [14]. TUDCA has positive effects in mouse models of protein-misfolding neurodegenerative diseases, including Alzheimer's disease (AD), Parkinson's disease, Huntington's disease, and amyotrophic lateral sclerosis (ALS) [15]. It is already used in humans for the treatment of primary biliary cirrhosis, and a phase III clinical trial in ALS is

under way [16]. We reasoned that TUDCA, by facilitating protein folding, could exert positive effects in SIL1-deficient cells, and eventually be used in combination therapy with PERK inhibitors [12].

Here we tested whether trazodone, DBM or TUDCA prevented or improved the motor phenotype and the associated UPR activation and PC degeneration in woozy mice. Since these drugs are already used in clinical practice, our hope was that evidence of therapeutic benefit in a preclinical model might support their repurposing for MSS.

## Materials and methods

### Study design

The aim of the study was to test whether drugs that partially inhibit the PERK branch of the UPR, or boost protein folding, are neuroprotective in the woozy mouse model of MSS. We therefore tested the effect of trazodone and DBM, which partially inhibit PERK signaling downstream of eIF2$\alpha$-P [13], and TUDCA, a chemical chaperone that protects MSS patient lymphoblastoid cells from ER stress [14].

Treatment was planned to overlap the initial phase of disease, starting from a presymptomatic stage (i.e. from 4 to 9 weeks of age), assessing the drugs' effects on early motor dysfunction and incipient PC degeneration, following our previously published experimental design [11] (Fig 1). Since we and others found no significant differences in motor performance and cerebellar anatomy between wild-type and heterozygous mice [6,11,17], we used heterozygous littermates of woozy mice as controls (referred to here as CT). No sex-related differences were observed in the woozy phenotype [11,17], so both male and female mice were included.

CT and woozy mice were trained in the beam walking and accelerated rotarod tests during the fourth week of life, and randomly assigned to treatments at the beginning of the fifth week. Motor performance was tested weekly and the experiments were terminated after five weeks (nine weeks of age). Performance in the beam walking test, which detects subtle deficits in motor skills and balance in the early phase of disease [11], was the primary endpoint. Based on the results of a previous experiment [11], assuming a mean number of missteps of 2, 6, 16, 23 and 37 in the woozy vehicle arm at 5, 6, 7, 8 and 9 weeks of age, an AR(1) correlation of 0.50 and residual variance of 2.00, six animals per arm are necessary to detect a 60% reduction of missteps (type I error: 0.05, power: 0.90). This target effect size was selected to align with the

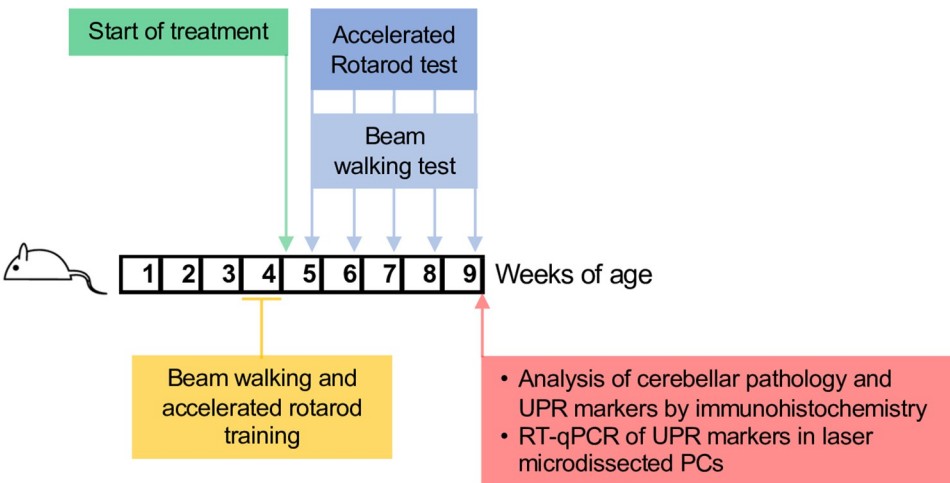

**Fig 1. Scheme of mouse treatment and analysis.**

reduction observed in previous studies using the PERK inhibitor GSK2606414. As secondary endpoint we also measured performance on the accelerating rotarod, which detects a significant deficit in woozy mice at nine weeks of age [11].

At the end of the treatment, brains were dissected. Half the brain was immersion-fixed for immunohistochemistry. The other half was snap-frozen for laser microdissection of PCs for RT-qPCR analysis of UPR markers [11].

**Cell culture.**   CHO-KI CHOP::luciferase cells [18] (a gift from David Ron, Cambridge University, UK) were cultured in Dulbecco's modified Eagle medium (DMEM)/F12(Ham) (Gibco) supplemented with 10% fetal bovine serum (Sigma Aldrich), 2 mM L-glutamine (Merck Millipore) and 1% penicillin/streptomycin (Merck Millipore) with 200 μg/mL G418 (Life Technologies). Cells were maintained at 37°C with 5% $CO_2$. Cells were regularly checked for mycoplasma contamination.

**CHOP::luciferase assay.**   CHO-KI cells stably transfected with CHOP::luciferase reporter [18] were cultured as described above: 0.6 x $10^5$ cells/$cm^2$ were plated in a 96-well plate and left growing for 24h. Cells were exposed to 3 μg/mL of tunicamycin or vehicle (dimethylsulphoxide, DMSO, Sigma), and treated for 6h with pharmacological compounds at concentrations ranging from 0.1 to 500 μM.

For all treatments the whole culture medium was replaced with fresh medium containing the compounds at the desired final concentrations, with or without addition of tunicamycin; control cells were treated by replacing the culture medium with fresh medium containing the vehicle. At the end of the treatment, cells were processed as follows. The cell medium was removed and cells were washed in PBS 1X three times before lysis in 100 μL of Glo Lysis Buffer (Promega) for 5 min at RT. Cell lysates were transferred to a black/clear bottom 96-well plate (Thermo Fisher Scientific) and 100 μL of Steady-Glo luciferase assay system (Promega) were added. After 5-min incubation, luciferase activity was measured using the GloMax 96 microplate luminometer (Promega).

## Mice

Woozy mice (CXB5/By-Sil1wz/J) [6] were obtained from The Jackson Laboratory (Stock No. 003777). They were maintained by heterozygous mating and genotyped by standard PCR, as recommended by the supplier. They were housed at controlled temperature (22 ± 2°C) with a 12/12-hour light/dark cycle and free access to pelleted food and water. The health and homecage behavior of the treated mice were monitored daily, according to guidelines for health evaluation of experimental laboratory animals [19].

Procedures involving animals and their care were conducted in conformity with the institutional guidelines at the IRCCS–Mario Negri Institute for Pharmacological Research in compliance with national (D.lgs 26/2014; Authorization no. 19/2008-A issued March 6, 2008 by Ministry of Health) and international laws and policies (EEC Council Directive 2010/63/UE; the NIH Guide for the Care and Use of Laboratory Animals, 2011 edition). They were reviewed and approved by the Mario Negri Institute Animal Care and Use Committee, which includes *ad hoc* members for ethical issues, and by the Italian Ministry of Health (Decreto no. 822/2020-PR). Animal facilities meet international standards and are regularly checked by a certified veterinarian who is responsible for health monitoring, animal welfare supervision, experimental protocols and review of procedures.

## Beam walking test

Mice were trained to walk along a metal beam 0.8 cm wide, 100 cm long, suspended 30 cm above bedding, for three days before testing. On the day of testing animals were given three

trials. Mice were video-recorded and the number of hindfoot missteps, falls and time to traverse the beam during the three trials were recorded by an investigator blinded to the experimental group. The average number of missteps and time to cross the beam during the three trials was used for statistical analysis.

### Accelerated rotarod test

The accelerating Rotarod 7650 model (Ugo Basile) was used as described [20]. Mice were trained three times before official testing. They were positioned on the rotating bar and allowed to become acquainted with the environment for 30 s. The rod motor was started at 7 rpm. and accelerated to 40 rpm. at a constant rate of 0.11 rpm/s for a maximum of 300 s. Performance was scored as latency to fall, in seconds. Animals were given three trials, and the average was used for statistical analysis.

### Immunohistochemistry

Mice were euthanized at the end of the treatments by cervical dislocation. For half of the animals from each experimental group brains were removed, post-fixed by immersion in 10% formalin and paraffin-embedded for immunoistochemical analysis. Serial sections (8 μm thick) were cut using a Leica microtome. Tissue sections were mounted on positively-charged glass slides (Superfrost Plus, Thermo Fisher Scientific), incubated at 58˚C for 20 min and then rehydrated by passing through an alcohol scale. An antigen retrieval step was done by washing samples in distilled water and boiling them in Antigen Decloaker Solution pH6 (BioOptica, diluted 1:10 in distilled $H_2O$) for three 5-min cycles in a microwave at 750 W power with 1 min of cooling between each cycle. Samples were then equilibrated at room temperature, washed with distilled water, incubated in 3% $H_2O_2$ for 15 min to inhibit endogenous peroxidase activity, then incubated for 1h at room temperature in blocking solution (10% FBS (Sigma), 5% bovine serum albumin (Sigma), 1% Triton X-100 (Sigma) in PBS 1X (Gibco)). Sections were incubated at 4˚C overnight with primary antibody rabbit anti-Calbindin D-28k (CB38, Swant, 1:2500) or rabbit poly-clonal anti GADD153/CHOP (sc-575, Santa Cruz Biothechnology, 1:250), diluted in blocking solution. Slides were then incubated with goat anti-rabbit Ig-G (H+L) Biotinylated secondary antibody (BA-1000, Vector Laboratories) 1:200 in blocking solution for 1h at room temperature, followed by 1h incubation with Avidin-Biotin Vectastain Elite ABC kit (Vector Laboratories) diluted 1:100 in PBS 1X. Finally, immunolabeling was visualized by application of the chromogen 3,3-diaminobenzidine (DAB) (Dako), followed by hematoxylin counterstaining (Mayer's hematoxylin, Sigma-Aldrich).

Images were acquired with a VS120 Virtual Slide Microscope by Olympus and analyzed with NIH ImageJ software. Four sagittal sections were quantified for each mouse cerebellum using an algorithm to segment the stained area. The calbindin immunostained area in the cerebellar molecular layer was expressed as positive pixels/total assessed pixels and reported as a percentage of the total stained area. CHOP-positive cells were counted over the entire perimeter of the Purkinje cell layer by an investigator blinded to the experimental group.

### Laser capture microdissection

Purkinje cells were isolated from the cerebellar molecular layer of woozy mice at nine weeks of age by laser capture microdissection (LCM). Samples were collected at the end of pharmacological treatment with either trazodone, DMB or TUDCA for all experimental groups. Serial sections (20 μm thick) were cut using a Leica cryostat and collected on metal frame slides (Leica) for LCM. Slices were left in 75% cold ethanol for 2 min, then washed twice for 1 min in MilliQ water, incubated 1 min in cresyl violet dye (1% of powder in absolute ethanol, Merck

Millipore), then held 5 sec in 90% ethanol, 5 sec in absolute ethanol, 1 min in fresh absolute ethanol, finally dried on absorbent paper and stored at -80°C with a dryer for later LCM. Before LCM, slices were equilibrated at -20°C, 4°C and room temperature for 15 min. PCs were microdissected from the cerebellar cortex with a laser microdissector (Leica LMD7).

## RT-qPCR

Total RNA was extracted from LCM PCs with the RNeasy Mini kit (Qiagen), and reverse-transcribed with a High-Capacity cDNA Reverse Transcription Kit (Applied Biosystems) using random primers. Quantitative PCR was done in Optical 96- or 384-well plates (Applied Biosystem) using the 7300 or 7900 Real Time PCR System (Applied Biosystems) and GoTaq qPCR Master Mix (Promega). Reaction conditions were 50°C for 2 minutes, 95°C for 10 min, then 95°C for 15 s alternating with 50°C for 1 min for 40 cycles, followed by 95°C for 15 s and 60°C for 1 min. The amplifications were always run in triplicate with a blank control consisting of a water template (RNase/DNase-free water).

Three housekeeping genes, HPRT1, GAPDH and SV2B, were used for data normalization. The primer pair sequences were: CCACCACACCTGAAAGCAGAA (Fw) and AGGTGCCCCCAA TTTCATCT (Rv) for CHOP; HPRT1 (Fw, CTTCCTCCTCAGACCGCTTTT and Rv, CATCATC GCTAATCACGACGC), GAPDH (Fw, TGTTTGTGATGGGTGTGAACC and Rv, CTGTGGTCAT GAGCCCTTCC) and SV2B (Fw, AGCATGTCACTGGCCATCAA and Rv, CCCAATCCCTATGCC TGAGAT).

## Drug treatments and bioanalysis

Trazodone (supplied by Angelini Pharma), DBM (dibenzoylmethane, Sigma Aldrich, D33454) and TUDCA (tauroursodeoxycholic acid sodium salt, Sigma Aldrich, T0266) were administered chronically according to dose regimens described previously [13,21,22]: trazodone 40 mg/kg ip daily; DBM 0.5% in the diet ad libitum; TUDCA either 0.4% in the diet or 500 mg/kg ip every three days. Normal diet and saline ip were given as vehicle controls. Trazodone and meta-chlorophenylpiperazine (m-CPP) concentrations were quantified in the plasma and brain homogenates using a HPLC/MS/MS system (Quattro Micro API triple quadrupole, Waters Corp., Manchester, UK) equipped with an electrospray ionization source (ESI) operating in positive ion mode. Trazodone-D6 and m-CPP-D6 were used as internal standards (IS). Chromatographic separation was achieved on an Alliance 2695 (Waters Corp.) with a Luna C18 column (Phenomenex Inc. USA) at 30°C, after liquid-liquid extraction. For quantification, eight-point calibration curves were generated by spiking 100 μL of control plasma or 500 μL brain homogenate (1g in 10 mL of $H_2O$) with trazodone or m-CPP, to final concentrations in the range of 0.02–5 μg/mL for plasma and 0.04–10 μg/g for brain. Quality control samples were prepared the same way to final concentrations of 0.075 μg/mL or 0.15 μg/g (low quality control, LQC), 0.3 μg/mL or 0.6 μg/g (mid-quality control, MQC) and 3.0 μg/mL or 6 μg/g (high quality control, HQC). and 0.03 nmol/mL for mCPP. Pharmacokinetic parameters were calculated using PKSolver, a freely available menu-driven add-in program for Microsoft Excel [23].

## Results

### TUDCA inhibits PERK pathway activation in CHOP::luciferase reporter cells

TUDCA is a chemical chaperone reported to promote protein folding and alleviate ER stress [15]. To verify its capacity to counteract ER stress, and specifically assess its effect on the PERK

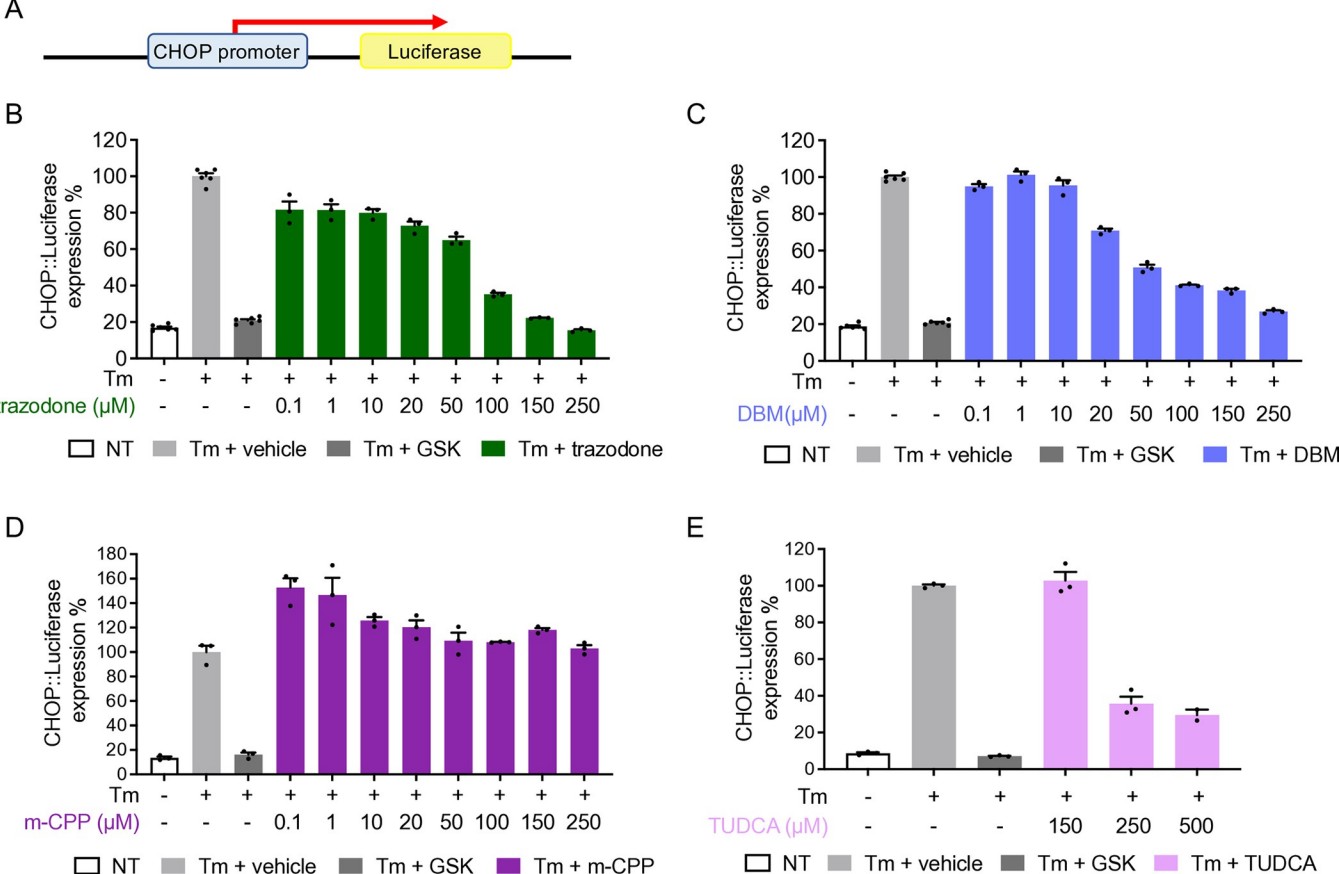

**Fig 2. Trazodone, DBM and TUDCA inhibits PERK pathway activation in CHOP::luciferase reporter cells in a dose-dependent manner.** A) CHOP:: luciferase cells carrying a reporter construct encoding the CHOP promoter driving a firefly luciferase transgene. B-E) Luciferase expression in CHOP:: luciferase cells treated with tunicamycin (Tm, red bar, 3 μg/mL) alone or with 20 μM GSK2606414 (GSK; dark gray), or trazodone (green), DBM (blue), m-CPP (purple) or TUDCA (pink) at the concentrations indicated. Data are the mean ± SEM of 3–6 replicates.

branch of the UPR, we used CHOP::luciferase cells which carry a reporter construct encoding the CHOP promoter and 5' UTR region driving a firefly luciferase transgene (Fig 2A). Exposure of CHOP::luciferase cells to the ER stressor tunicamycin induces PERK-mediated eIF2α phosphorylation which stimulates the selective translation of ATF4, which in turn induces robust luciferase expression [18]. Thus, we measured the ability of TUDCA to attenuate the luciferase response to tunicamycin in CHOP::luciferase cells. Trazodone and DBM were tested in parallel, as their eIF2α-P signaling inhibitory activity had been characterized in these very same cells [13]. In addition, we tested m-CPP, the main trazodone metabolite to see whether it may contribute to trazodone-mediated inhibition of eIF2α-P signaling in vivo (see below). CHOP::luciferase cells were exposed to tunicamycin and the vehicle (DMSO) or the test compounds at concentrations ranging between 0.1 and 500 μM, and luciferase activity was measured after 6h, as described [13]. Consistent with previous findings [13], trazodone and DBM dose-dependently inhibited tunicamycin-induced luciferase expression in the reporter cell line, with $IC_{50}$ of 80 and 70 μM, respectively, whereas m-CPP had no effect up to 250 μM (Fig 2B–2D). TUDCA also inhibited the luciferase response at concentrations $\geq$ 250 μM (Fig 2E).

**Trazodone slightly improves the performance of woozy mice in the beam walking test, with no effect on PERK/eIF2α signaling and PC degeneration.** Woozy mice develop ataxia first detectable in the beam walking test at seven weeks of age, and on the accelerating rotarod

at nine weeks, at which stage immunohistochemistry finds increased numbers of CHOP-positive PCs and a significant reduction in PC bodies and dendritic arborizations [11].

We assessed the effects of trazodone on early motor dysfunction and PC degeneration, following our previously published experimental design so we could directly compare the efficacy of trazodone with GSK2606414 [11] (Fig 1). CT and woozy mice were trained in the beam walking and accelerated rotarod tests in the fourth week of life, and treatment was started at the beginning of the fifth week. Trazodone was given 40 mg/kg i.p. daily, a dosing regimen previously shown to prevent neurodegeneration and clinical disease in prion-infected mice, and to reduce hippocampal degeneration in the rTg4510 $tau_{P301L}$ mouse model of frontotemporal dementia [13]. Saline ip was given as control. Motor performance was tested weekly and the experiment was terminated after five weeks (nine weeks of age) for analysis of CHOP expression and PC degeneration.

There were no differences in beam walking performance between the different groups of mice up to six weeks of age (Fig 3A and 3B). From seven weeks, the number of hindfoot missteps and the time to traverse the beam increased significantly in vehicle-treated woozy mice compared to controls. Trazodone-treated woozy mice performed better, making significantly fewer missteps than vehicle treated woozy mice (Fig 3A).

However, trazodone had no effect on rotarod performance. At five weeks of treatment, the latency to fall of both vehicle- and trazodone-treated woozy mice was similarly reduced compared to CT mice (Fig 3C).

Immunohistochemistry with anti-calbindin antibody for quantitative assessment of PCs found similar reductions in the percentages of the calbindin-positive area in vehicle- and trazodone-treated woozy mice compared to controls (Fig 4A and 4B). Immunostaining with anti-CHOP antibody found no reduction of CHOP-positive PCs in trazodone-treated woozy mice (Fig 4C and 4D).

RT-qPCR analysis of laser capture microdissected PCs indicated a 3.6-fold increase in CHOP mRNA in vehicle-treated woozy mice compared to vehicle-treated CT ones (Fig 5). CHOP mRNA levels in vehicle- and trazodone-treated woozy mice were similar, consistent with the lack of effect of the drug on PERK/eIF2α signaling seen by CHOP immunohistochemistry (Fig 5).

Taken as a whole, these results indicate that trazodone has a limited beneficial effect on motor behavior assessed by beam walking but not by the rotarod test, and this is not due to neuroprotective inhibition of PERK/eIF2α signaling.

**Pharmacokinetics of trazodone in woozy mice.** We wondered whether the lack of effect of trazodone was due to an altered pharmacokinetic (PK) profile in woozy mice, which have a genetic background (CXB5/ByJ) different from that of mice in which trazodone had neuroprotective effects (Sv129/C57BL/FVB or C57BL/6J) [13,24]. Plasma and brain levels of trazodone were measured in groups of CXB5/ByJ and C57BL/6J mice (3–4) given 40 mg/kg trazodone ip 0.5, 1, 2, 3, 4 and 6h post-dosing. The levels of the main trazodone metabolite m-CPP were also measured.

Trazodone levels were the highest at the first time-point (30 min) in both plasma and brain (Fig 6). No major differences were observed in the PK profile of trazodone between CXB5/ByJ and C57BL/6J mice, except for slightly higher concentrations at the early time points and slightly faster elimination from the plasma of CXB5/ByJ mice; results were similar in the brain (Fig 6). The brain to plasma ratio calculated on the $AUC_{0-t}$ was similar in the two mouse strains (1.62 and 1.47 in C57BL/6J and CXB5/ByJ mice).

Trazodone metabolism to m-CPP was slightly impaired in CXB5/ByJ mice compared to C57BL/6J mice. This was evident from the m-CPP/trazodone ratio, calculated on the $AUC_{0-t}$ in the plasma, which was 0.34 in CXB5/ByJ and 0.67 in C57BL/6J mice. Similarly to trazodone, m-CPP elimination from the plasma was slightly faster in CXB5/ByJ than C57BL/6J mice.

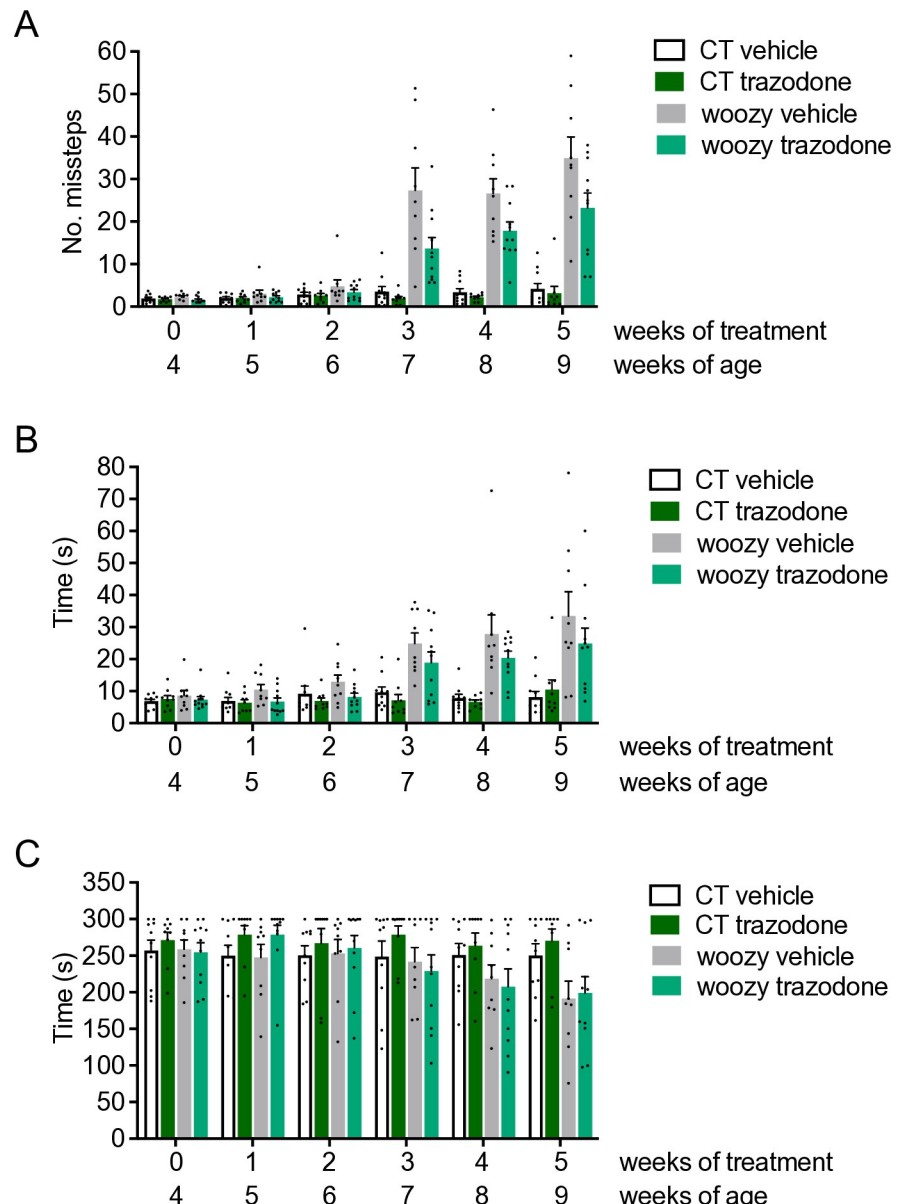

**Fig 3. Trazodone slightly ameliorates motor performance of woozy mice on the beam walking but not rotarod test.** (A) Groups of CT and woozy mice were given trazodone or the vehicle starting from four weeks of age, and tested weekly for their ability to walk on a suspended beam. Each mouse was tested three times and the mean number of hindfoot missteps was scored. Data are the mean ± SEM of ten vehicle- and nine trazodone-treated CT, nine vehicle- and eleven trazodone-treated woozy mice. $F_{1,17}$ = 34.65, p<0.0001 CT vehicle *vs.* woozy vehicle; $F_{1,18}$ = 7.59, p = 0.013 woozy trazodone *vs.* woozy vehicle by two-way analysis of variance for repeated measures (RM ANOVA). (B) Mean time to traverse the beam during the three trials. Bars indicate the mean ± SEM of time (s). $F_{1,17}$ = 13.71, p = 0.0018 CT vehicle *vs.* woozy vehicle; $F_{1,18}$ = 3.03, p = 0.098 woozy trazodone *vs.* woozy vehicle by two-way RM ANOVA. (C) Groups of CT and woozy mice were tested on the rotarod at the ages indicated. Each mouse was tested three times, and the mean latency to fall was calculated. Bars indicate the mean ± SEM of latency to fall (s). $F_{1,35}$ = 10.47, p = 0.0026 CT *vs.* woozy at nine weeks of age by two-way ANOVA.

m-CPP crosses the blood-brain barrier (BBB) much better than the parent compound, with brain-to-plasma ratios of 30.9 in C57BL/6J and 20.1 in CXB5/ByJ mice, suggesting reduced BBB permeability in the latter. As a consequence, and the reduced metabolism of trazodone to

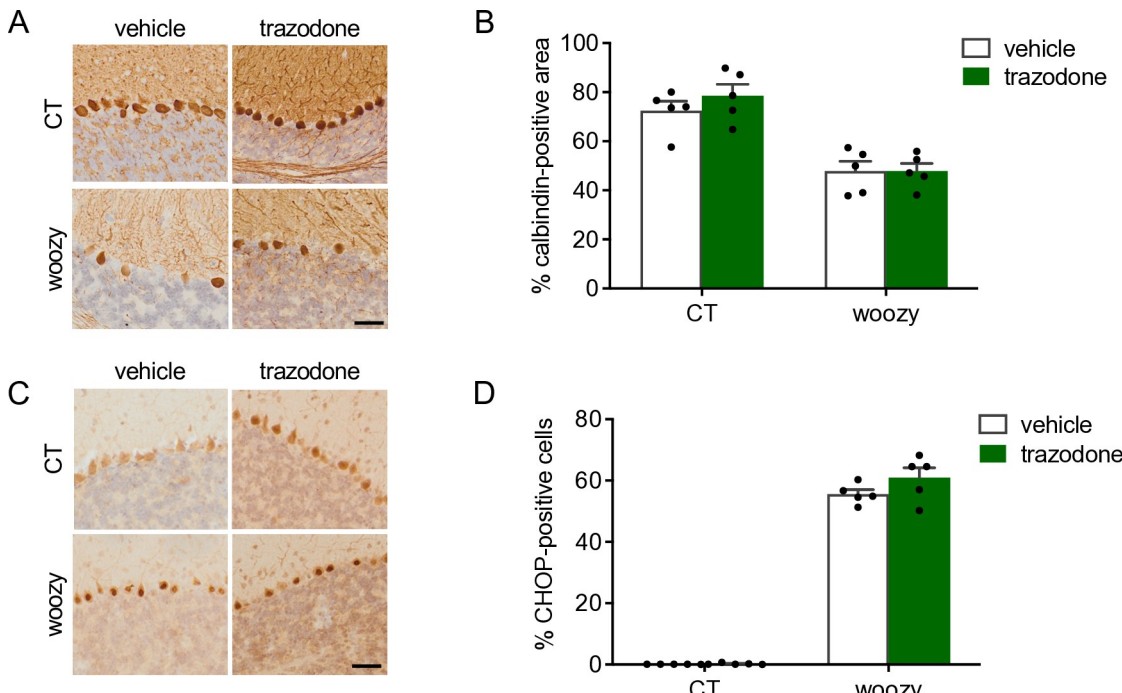

**Fig 4. Trazodone does not prevent PERK signaling activation and degeneration of Purkinje cells.** (A) Calbindin immunostaining in cerebellar sections of CT and woozy mice treated with trazodone or the vehicle for five weeks. (B) Percentage of the calbindin positive area in the cerebellar cortex. Data are the mean ± SEM of three brain sections from five animals per experimental group. (C) CHOP immunostaining in cerebellar sections of CT and woozy mice treated with trazodone or the vehicle for five weeks. (D) The number of CHOP-positive PC was analyzed by immunohistochemistry and expressed as the percentage of total PCs. Data are the mean ± SEM of three brain sections from five mice per experimental group.

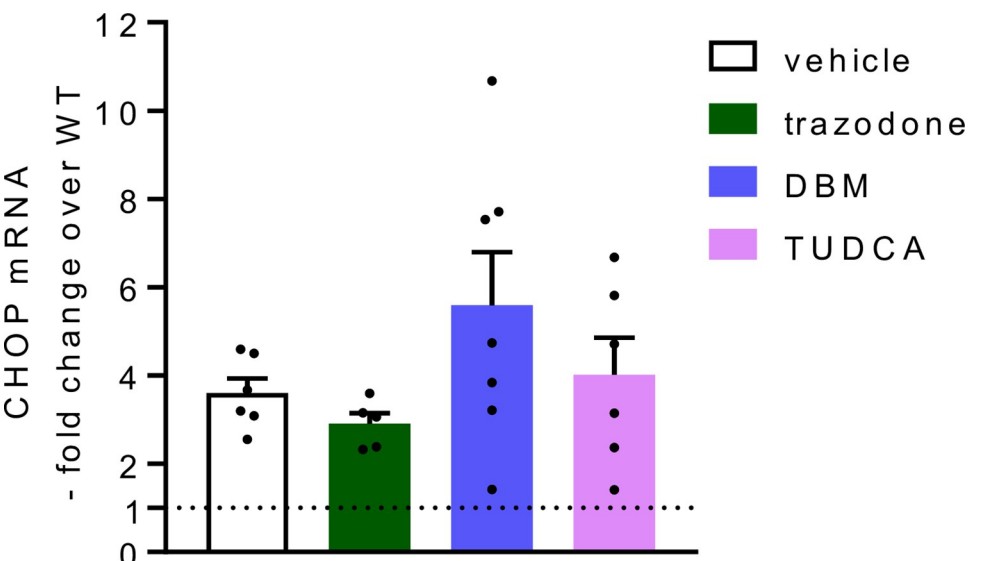

**Fig 5. CHOP mRNA in the cerebellum of woozy mice.** Total RNA was extracted from Purkinje cells laser-microdissected from the cerebella of six vehicle-, five trazodone-, seven DBM- and six TUDCA-treated woozy mice (data are the mean ± SEM). RNA was reverse-transcribed and analyzed by RT-qPCR. mRNAs were quantified by the ΔΔCt method and expressed as the -fold difference from the levels in age-matched CT mice (dotted line).

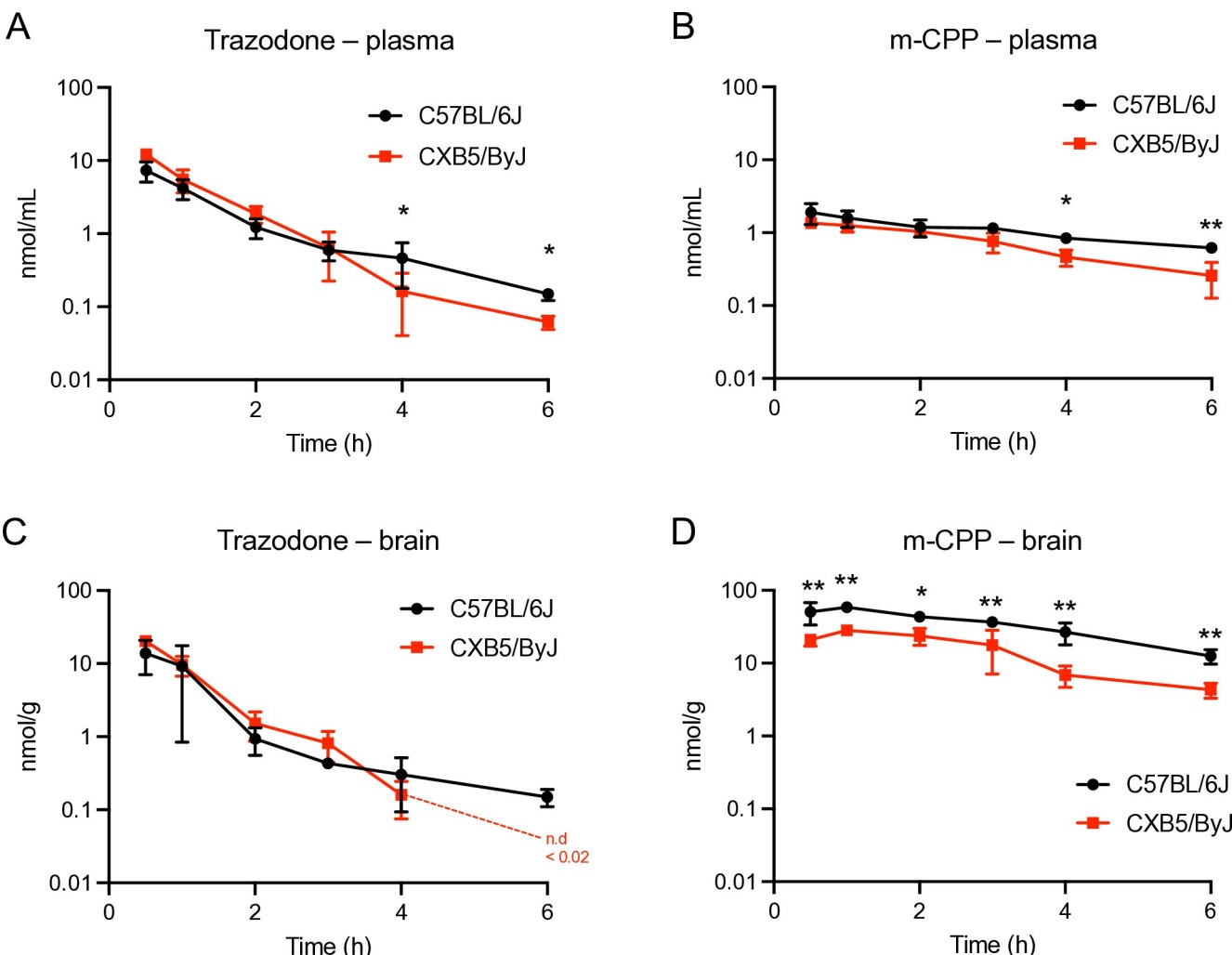

**Fig 6. Comparison of trazodone and m-CPP pharmacokinetics in CXB5/ByJ and C57BL/6J mice.** Mean plasma and brain concentration-times of trazodone (A, C) and its metabolite m-CPP (B, D) after ip injection of 40 mg/kg trazodone in C57BL/6J (black symbols) and CXB5/By/J mice (red symbols). Each point is the mean ± SD of 2–4 mice. * $p < 0.05$, ** $p < 0.01$ by two-way ANOVA, Sidak's multiple comparison test.

m-CPP, the brain levels of m-CPP were consistently significantly lower in CXB5/ByJ than C57BL/6J mice, with $AUC_{0-t}$ respectively 87.9 (95%CI: 64–101) and 196 (95%CI: 164–215). The elimination rates of m-CPP from the brain were similar in C57BL/6J and CXB5/ByJ mice.

Trazodone and m-CCP levels were also measured in the CT and woozy mice that had been chronically treated ip for five weeks and culled 6h after the last dose (experiment shown in Fig 3). These mice have the CXB5/ByJ background but carry either one (CT) or two (woozy) copies of the mutant $Sil1^{wz}$ allele (see Study Design in the Material and Methods). Trazodone and m-CPP levels were similar in the two groups of mice (Fig 7), indicating that loss of SIL1 function had no effect on the drug pharmacokinetics and metabolism. The drug levels in chronically treated mice were comparable to those found 6h after acute treatment (see CXB5/ByJ group in Fig 6) indicating no accumulation of trazodone or m-CPP, in line with their short half-lives. These data also indicate no significant metabolic induction after chronic treatment.

**DBM and TUDCA do not improve motor performance and cerebellar pathology in woozy mice.** Groups of CT and woozy mice were given regular chow (vehicle) or chow

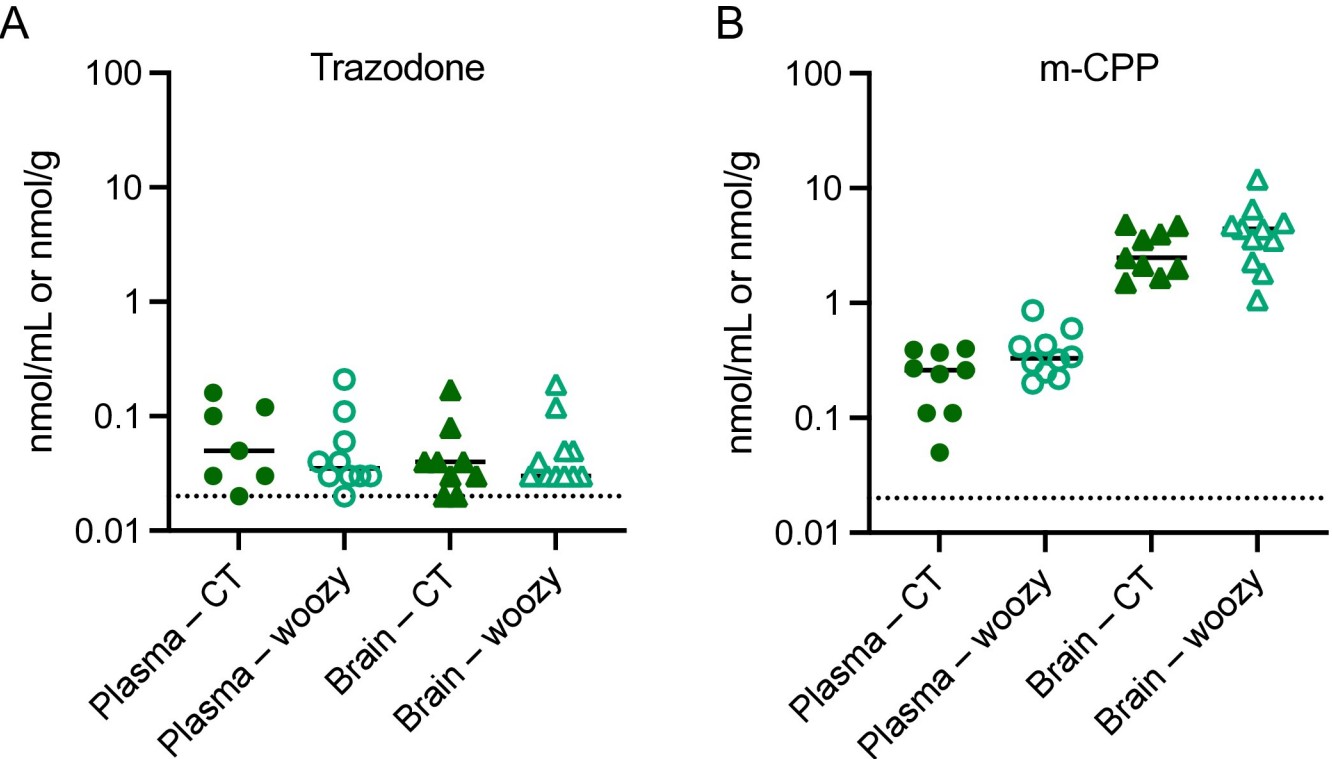

**Fig 7. Trazodone and m-CPP concentrations in the plasma and brain of CT and woozy mice after chronic treatment with trazodone.** Concentrations of trazodone (A) and m-CPP (B) in the plasma and brain were measured 6h after the last dose of five-week ip treatment with 40 mg/kg/day trazodone. Animals are the same as in Fig 3.

supplemented with 0.5% DBM or 0.4% TUDCA starting from the fourth week of age. Motor function was assessed weekly in the beam walking and rotarod tests for five weeks. Neither treatment had any effect on the development of woozy motor dysfunction, as evidenced by both tests (Fig 8).

Immunohistochemistry of the cerebellum with anti-calbindin and anti-CHOP antibodies, and CHOP mRNA levels in laser microdissected PCs in mice culled at the end of the treatment indicated no protection from neurodegeneration or inhibition of PERK/eIF2α signaling (Figs 5, 9A and 9B).

TUDCA is generally given orally in the diet [21,25–27]. However, there is also evidence of neuroprotective activity in mice dosed ip from 50 to 500 mg/kg which can increase bile acid brain levels ~7-fold [22,28–31]. Therefore we ran an additional experiment, treating mice ip with 500 mg/kg TUDCA every three days, a regimen previously shown to attenuate amyloid-β pathology and reduce neuroinflammation and synaptic loss in the APP/PS1 mouse model of AD [22]. Treatment was started at four weeks of age and motor behavior was assessed weekly by the beam walking test. There was a clear age-dependent increase in the number of missteps and the time to traverse the beam between woozy and CT mice, but no difference between vehicle- and TUDCA-treated woozy mice (Fig 10).

## Discussion

We previously found that PERK pathway activation in Purkinje cells preceded cerebellar degeneration and ataxia in woozy mice [11]. Pre-symptomatic treatment with the PERK inhibitor GSK2606414 to allow full translational recovery delayed Purkinje cell death, prolonging

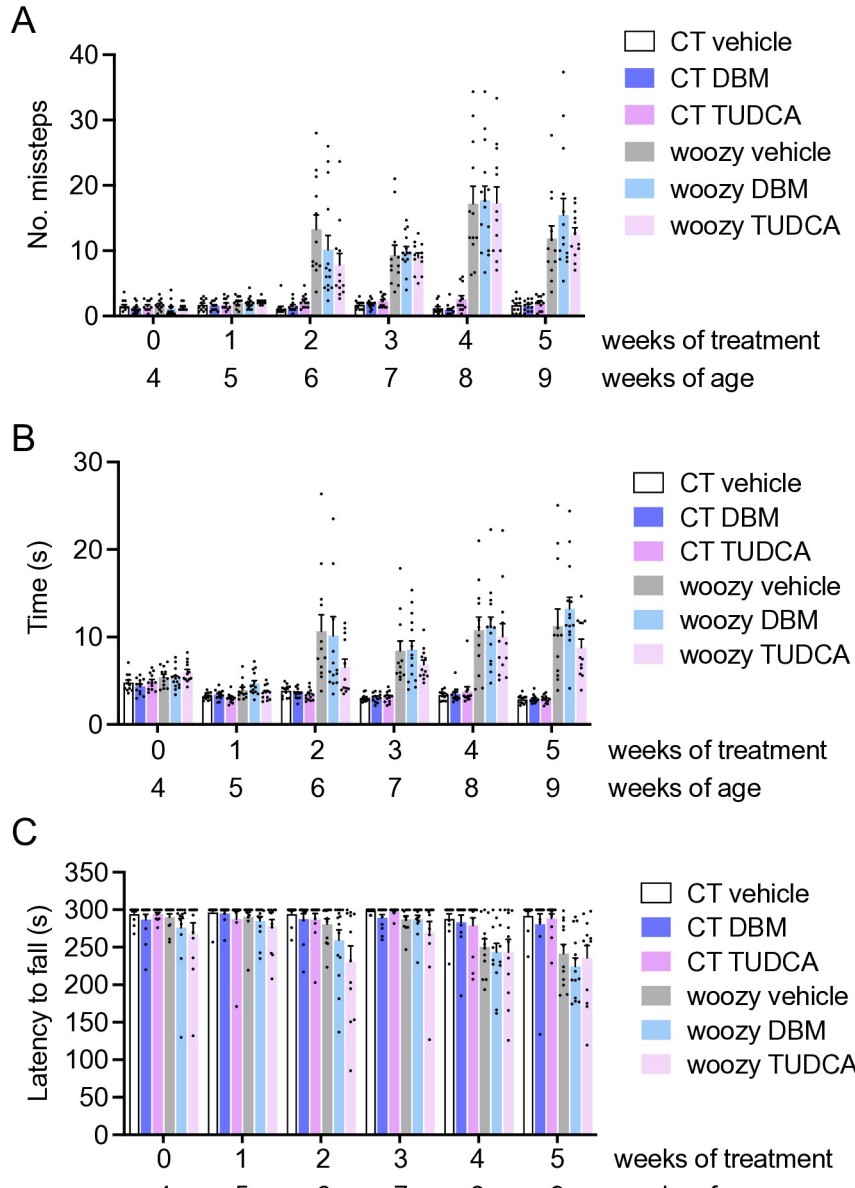

**Fig 8. DBM and TUDCA do not improve the motor performance of woozy mice.** (A) Groups of CT and woozy mice were given DBM, TUDCA or the vehicle starting from four weeks of age, and tested weekly for their ability to walk on a suspended beam. Each mouse was tested three times and the mean number of hindfoot missteps was scored. Data are the mean ± SEM of 12 vehicle-, 12 DBM- and 12 TUDCA-treated CT, 12 vehicle-treated woozy, 14 DBM- and 12 TUDCA-treated woozy mice. (B) Mean time to traverse the beam in the three trials. Bars indicate the mean ± SEM of time (s). (C) Mice were tested on the rotarod at the ages indicated. Each mouse was tested three times, and the mean latency to fall was calculated. Bars indicate the mean ± SEM of latency to fall (s).

the asymptomatic phase of the disease [11]. In the present study, we tested trazodone and DBM, which partially inhibit PERK signaling downstream of eIF2α-P, offering neuroprotective effects similar to GSK2606414 but without pancreatic toxicity [13]. We hypothesized that partial translational recovery by trazodone or DBM might even provide better neuroprotection than GSK2606414 by preventing overloading the inefficient ER folding machinery of SIL1-deficient Purkinje cells [12]. We also tested TUDCA, a chemical chaperone previously shown to

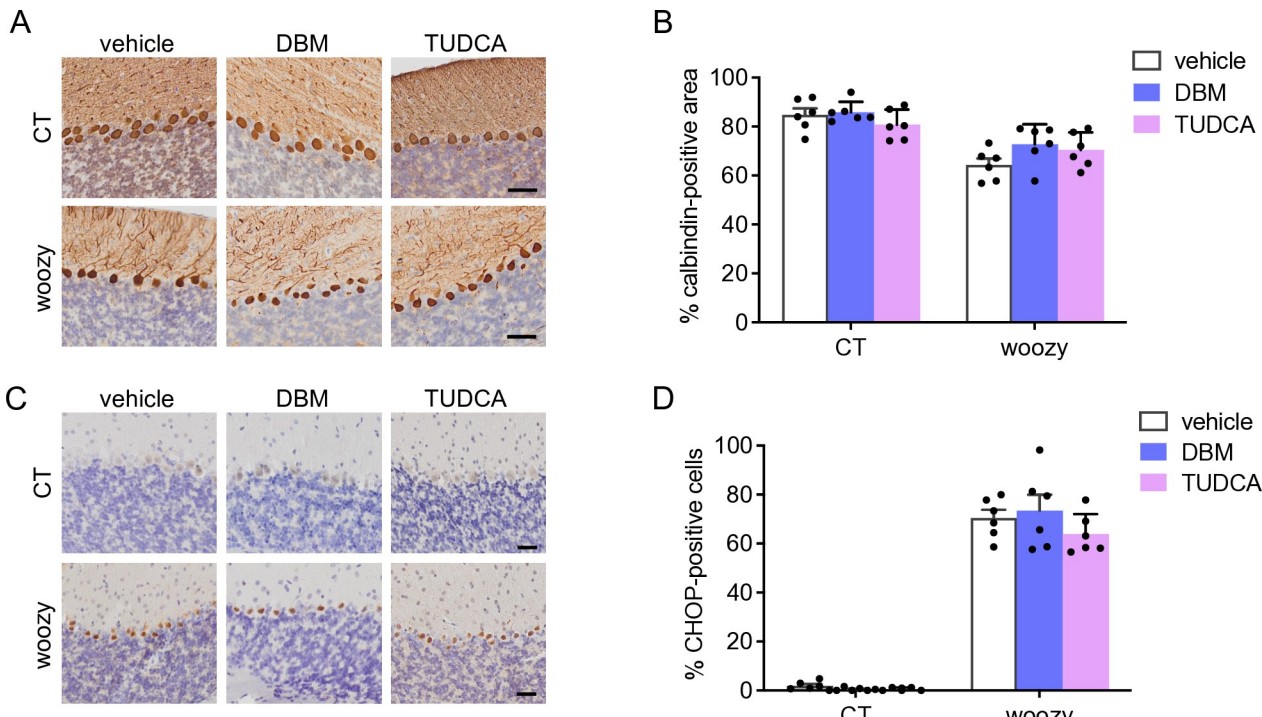

**Fig 9. DBM and TUDCA given in diet do not prevent PERK signaling activation and degeneration of Purkinje cells in woozy mice.** (A) Calbindin immunostaining in cerebellar sections of CT and woozy mice treated with DBM, TUDCA or the vehicle for five weeks, scale 40 μm. (B) Percentage of the calbindin positive area in the cerebellar cortex. Data are the mean ± SEM of 1–3 brain sections from six mice per experimental group. (C) CHOP immunostaining in cerebellar sections of CT and woozy mice treated with DBM, TUDCA or the vehicle for five weeks, scale 20 μm. (D) The number of CHOP-positive PC was analyzed by immunohistochemistry and expressed as the percentage of total PCs. Data are the mean ± SEM of 1–3 brain sections from six animals per experimental group.

alleviate ER stress-induced toxicity in a MSS cell model, as a potential candidate for combination therapy with PERK inhibitors to enhance neuroprotection [12]. Since trazodone, DBM and TUDCA are suitable for clinical use, positive outcomes in the woozy mouse model might have supported their repositioning for the treatment of MSS. However, we found these compounds had no effects on clinical, histopathological and molecular readouts, negating their potential for repurposing. Additionally, the observation that trazodone and DBM failed to inhibit PERK signaling in woozy mice does not support their broad applicability to neurodegenerative diseases involving maladaptive PERK activation.

Pharmacological inhibition of the PERK branch of the UPR has emerged as a promising therapeutic strategy for protein misfolding neurodegenerative diseases [10]. However, experimental PERK pathway inhibitors active in preclinical models, such as GSK2606414 or the integrated stress response inhibitor ISRIB, face limitations for clinical translation due to their pancreatic toxicity or unfavorable pharmacokinetics [32,33]. In a repurposed drug screening trazodone and DBM partially inhibited PERK signaling downstream of eIF2α-P [13]. These compounds offered neuroprotective effects while preserving a level of eIF2α-P-mediated translational repression essential for pancreatic secretory function, making them potential candidates for clinical application. Of particular interest is trazodone, a serotonin antagonist and reuptake inhibitor, which has undergone extensive trials since the 1970s for conditions like major depressive disorder, with a wealth of safety data supporting its use.

Here we confirmed the ability of trazodone and DBM to inhibit PERK signaling in CHOP:: luciferase cells exposed to the ER stressor tunicamycin. However, compared to results of

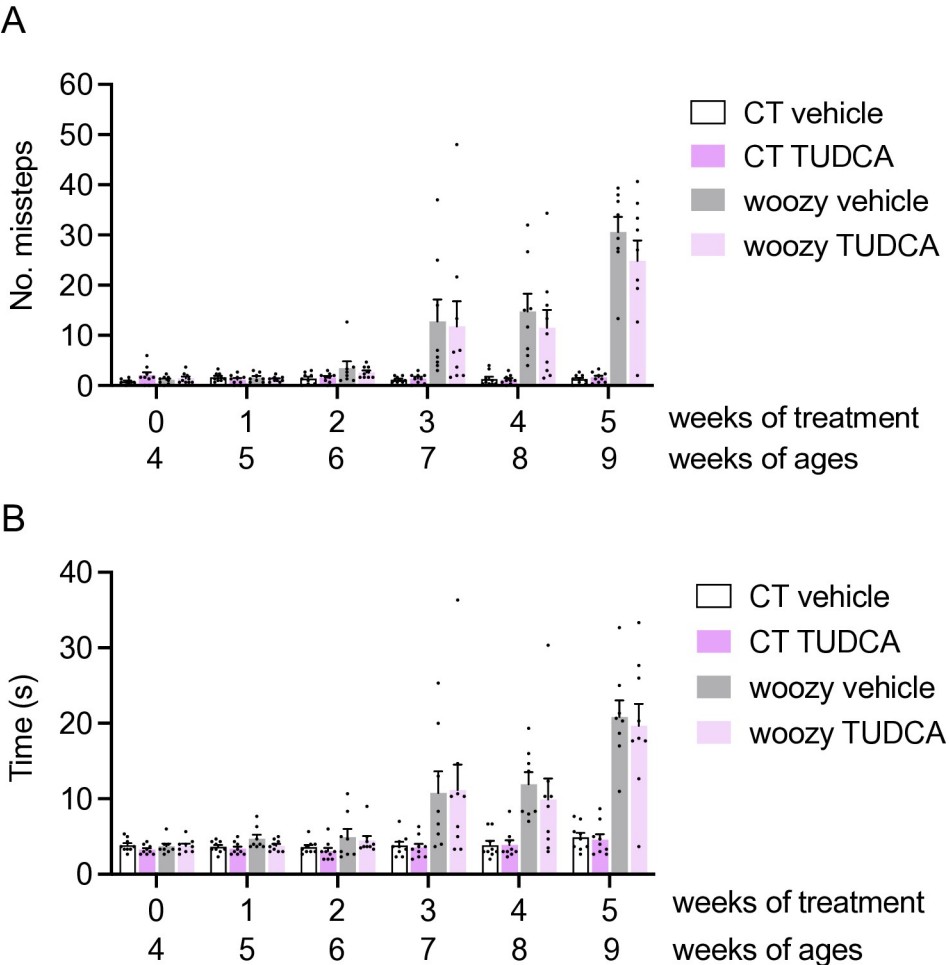

**Fig 10. TUDCA ip does not ameliorate beam walking performance of woozy mice.** (A) Groups of CT and woozy mice were given TUDCA or the vehicle starting from four weeks of age, and tested weekly for their ability to walk on a suspended beam. Each mouse was tested three times and the mean number of hindfoot missteps was scored. (B) Mean time to traverse the beam during the three trials. Data are the mean ± SEM of nine vehicle- and nine TUDCA-treated CT, eight vehicle- and nine TUDCA-treated woozy mice.

Halliday et al. using the same cell line [13], higher concentrations were needed to achieve similar inhibitory effects. Moreover, the inhibition did not reach a plateau, not supporting the argument that their lower pancreatic toxicity compared with GSK2606414 relates to some intrinsic limitation of PERK signaling inhibitory activity [13].

We found that TUDCA attenuated the tunicamycin-induced luciferase response in CHOP:: luciferase cells, similarly to trazodone and DBM. This quite likely reflects the chemical chaperone's ability to enhance protein folding and reduce ER stress, rather than directly interfering with PERK signaling [14,15]. Thus with different mechanisms all three compounds attenuated ER stress-activated signaling in cultured cells, supporting their testing in vivo.

The CHOP::luciferase assay validate the compounds' activity at the molecular level. However, translating results in this cell model to the in vivo setting in terms of predictive efficacy and drug dosing is challenging. Thus we relied on previous pharmacokinetics studies and dosing regimens shown to be effective in other neurodegenerative mouse models [13,21,24–26,28–31].

Pre-symptomatic treatment of woozy mice with trazodone, DBM or TUDCA, following the same experimental design that demonstrated the efficacy of GSK2606414 [11], indicated no beneficial disease-modifying effects. There were no differences in the onset and development of motor dysfunction, assessed in the beam walking and rotarod tests, or incipient cerebellar pathology evaluated by quantitative calbindin immunostaining, between woozy mice treated with vehicle or the test compounds.

Only trazodone slightly improved woozy performance in the beam walking task. However, this was not associated with inhibition of PERK signaling, as confirmed by quantitative analysis of CHOP-immunopositive PCs and CHOP mRNA expression levels in laser capture microdissected PCs. Thus, in stark contrast with the marked rescue of early motor deficits and neurodegeneration, and the reduced CHOP activation seen in woozy mice treated with GSK2606414, the better motor performance of trazodone-treated mice was not related to neuroprotective inhibition of PERK signaling.

Serotonergic neurotransmission modulates neuronal excitability and synaptic plasticity in the cerebellum [34,35]. Conceivably therefore trazodone may have enhanced cerebellar function in woozy mice by raising serotonin levels or modulating serotonin receptor activity, compensating for the loss of Purkinje neurons and improving motor coordination and balance. Additional experiments, such as in vivo microdialysis, and analysis of serotonin receptor expression and activity in the cerebellum of trazodone-treated woozy mice, would be necessary to test this hypothesis.

Woozy mice were treated with trazodone, DBM or TUDCA according to dose regimens and administration routes that have proved effective in a wide range of neurodegenerative disease mouse models, including prion-diseased mice, and transgenic rTg4510 $tau_{P301L}$ and APP/PS1 mice, modeling frontotemporal dementia and AD [13,21,24–27]. Therefore it is unlikely that their lack of effect was due to insufficient bioavailability.

One possible explanation could be altered drug pharmacokinetics in woozy mice, potentially leading to lower bioactive concentrations in the CNS. To explore this we compared trazodone pharmacokinetics between mice with the woozy genetic background (CXB5/ByJ) and C57BL/6J mice, as trazodone had previously demonstrated neuroprotective PERK signaling inhibitory activity in the C57BL/6J background [24]. Results indicated no major differences. Although trazodone elimination was slightly faster in CXB5/ByJ mice, this was not due to an accelerated metabolism to its main metabolite, m-CPP, which was mildly impaired in these mice. Consistently, CXB5/ByJ mice also showed a slightly faster elimination of mCPP.

Brain levels of trazodone were very similar between CXB5/ByJ and C57BL/6J mice, whereas the brain levels of mCPP were 2-fold lower in CXB5/ByJ compared to C57BL/6J mice. Lower m-CPP brain levels might have explained the lack of PERK signaling inhibition if m-CPP was the main mediator of trazodone activity. However, we found no evidence of PERK inhibitory activity of m-CPP in CHOP::luciferase cells.

Furthermore, after chronic treatment with trazodone, we found no differences in brain and plasma levels of trazodone and mCPP between CT and woozy mice, indicating that the pharmacokinetics of trazodone was not influenced by loss of SIL1 function. Overall, these data suggest that the lack of trazodone's PERK signalling inhibitory activity in woozy mice is unlikely due to pharmacokinetic reasons and insufficient trazodone brain levels.

Our findings that trazodone and DBM have no neuroprotective PERK signalling inhibitory activity in woozy mice do not support their repositioning for the treatment of MSS. They also suggest that these compounds may not be broadly useful for fine-tuning PERK signalling in neurodegeneration. An alternative approach to inhibit PERK signalling without systemic toxicity is to reduce eIF2α binding to the PERK kinase domain by enhancing Akt-mediated phosphorylation of PERK threonine (Thr) residue 799 [36]. It may be interesting to see whether

induction of Thr$^{799}$ phosphorylation with a small-molecule activator of Akt reduces PERK signaling and prevents cerebellar degeneration in woozy mice.

The lack of disease modifying activity of TUDCA in woozy mice is disappointing, given the broad beneficial activity of this compound in a number of preclinical models of neurodegenerative disease [37], and evidence of protection in MSS patients' cells [14]. However, there have been other instances where protective effects of TUDCA in vitro were not confirmed in vivo. For example, while TUDCA reduced prion replication and toxicity in cells and cultured organotypic cerebellar slices, pre-symptomatic treatment of prion-infected mice with TUDCA at 0.4% or 1% in the diet did not prolong their survival [38,39].

## Conclusions

Our study aimed to explore potential therapeutic strategies targeting the PERK pathway or aiding ER protein folding in MSS. Despite promising outcomes reported in previous research, this investigation into the efficacy of trazodone, DBM, and TUDCA found no disease-modifying effects. Future research may explore other pharmacological strategies for non-toxic PERK signaling inhibition, such as modulation of physical interaction between PERK and eIF2α [36]. Additionally, in the light of evidence that the reintroduction of a single wild-type *Sil1* allele, or transgenic expression of the SIL1 homologous 150-kDa oxygen-regulated protein (ORP150, also known as GRP170), fully rescues the woozy mouse phenotype [6,7], gene therapy approaches aimed at restoring SIL1 function may be another promising avenue worth pursuing.

## Supporting information

**S1 File.**
(XLSX)

## Acknowledgments

We thank David Ron for providing the CHO-KI cells stably transfected with the CHOP::luciferase reporter, and Angelini Pharma for providing trazodone. We are grateful to Stefano Fumagalli for advice on quantification of calbindin and CHOP immunostaining.

## Author Contributions

**Conceptualization:** Roberto Chiesa.

**Formal analysis:** Giada Lavigna, Marco Gobbi.

**Funding acquisition:** Roberto Chiesa.

**Investigation:** Giada Lavigna, Anna Grasso, Chiara Pasini, Valentina Grande, Laura Mignogna, Elena Restelli, Antonio Masone, Claudia Fracasso, Jacopo Lucchetti.

**Supervision:** Marco Gobbi, Roberto Chiesa.

**Writing – original draft:** Giada Lavigna, Roberto Chiesa.

**Writing – review & editing:** Marco Gobbi, Roberto Chiesa.

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
