## [Decision Letter · Decision Letter 0]

1 Dec 2024

PONE-D-24-30565Trazodone, dibenzoylmethane and tauroursodeoxycholic acid do not prevent motor dysfunction and neurodegeneration in Marinesco-Sjögren syndrome micePLOS ONE

Dear Dr. Chiesa,

Thank you for submitting your manuscript to PLOS ONE. After careful consideration, we feel that it has merit but does not fully meet PLOS ONE’s publication criteria as it currently stands. Therefore, we invite you to submit a revised version of the manuscript that addresses the points raised during the review process.

We look forward to receiving your revised manuscript.

Kind regards,

Shahbaz Ahmad Zakki

Academic Editor

PLOS ONE

“European Union:Roberto Chiesa PNRR-MR1-2022-12375730; Fondazione Telethon (FT):Roberto Chiesa GGP20071”

“We thank David Ron for providing the CHO-KI cells stably transfected with the CHOP::luciferase

reporter, and Angelini Pharma for providing trazodone. We are grateful to Stefano Fumagalli for

advice on quantification of calbindin and CHOP immunostaining. This work was funded by

Fondazione Telethon (GGP20071). Maintenance of the woozy mouse colony was partially supported

by the European Union - Next Generation EU - NRRP M6C2 - Investment 2.1 Enhancement and

strengthening of biomedical research in the NHS (PNRR-MR1-2022-12375730”

“European Union:Roberto Chiesa PNRR-MR1-2022-12375730; Fondazione Telethon (FT):Roberto Chiesa GGP20071”

Reviewers' comments:

Reviewer's Responses to Questions

**Comments to the Author**

1. Is the manuscript technically sound, and do the data support the conclusions?

Reviewer #1: Yes

Reviewer #2: Yes

2. Has the statistical analysis been performed appropriately and rigorously? 

Reviewer #1: Yes

Reviewer #2: Yes

3. Have the authors made all data underlying the findings in their manuscript fully available?

Reviewer #1: Yes

Reviewer #2: Yes

4. Is the manuscript presented in an intelligible fashion and written in standard English?

Reviewer #1: Yes

Reviewer #2: Yes

5. Review Comments to the Author

Reviewer #1: The target article uses advanced methods to investigate potential treatments for Marinesco-Sjogren Syndrome (MSS). Despite testing several drugs and conducting extensive behavioral and histochemical analysis, the treatment failed to show disease modifying effects. This highlights the challenges of translating neuroprotective strategies. However, this study represents a significant step forward in understanding the limitations of current pharmacological approaches and points towards the exploration of alternative strategies for more effective interventions.

Reviewer #2: General Comment:

The manuscript provides a solid and methodologically sound investigation of the effects of trazodone, dibenzoylmethane (DBM), and tauroursodeoxycholic acid (TUDCA) on motor dysfunction and neurodegeneration in a preclinical mouse model of Marinesco-Sjögren syndrome (MSS). The study is comprehensive, well-executed, and the claims made are reasonable. However, there are some minor areas that require clarification and refinement to improve the manuscript's clarity and impact.

Specific Comments:

Comment #1:

The abstract is detailed but could be made clearer or more focused. I suggest emphasizing the current primary findings, specifically the lack of neuroprotective effects, and discussing their potential implications for future MSS research if possible. This would provide readers with a clearer and more focused overview of the study’s outcomes.

Comment #2:

I suggest the author to provide more context for selecting trazodone, DBM, and TUDCA as the focus of this study. Expanding on the rationale behind testing these compounds specifically in MSS, as well as discussing the role of the PERK pathway in MSS pathophysiology, would strengthen the manuscript. It would also help to explain why these drugs were hypothesized to be effective in this context.

Comment #3:

While the manuscript includes a power analysis, it would benefit from a more detailed justification for selecting the 60% reduction in missteps as the target effect size. Clarifying this decision would improve the transparency and credibility of the study’s design. Additionally, further discussion on the rationale for the chosen dosing regimens particularly in relation to previous studies or pharmacokinetic data would provide a clearer foundation for the experimental approach.

The manuscript could also clarify the relevance of the CHOP::luciferase assay to the in vivo findings, especially for readers who may be less familiar with molecular techniques. Finally, I encourage the authors to emphasize the translational challenges identified in the study and offer recommendations for future drug testing in MSS models.

Comment #4:

I recommend expanding the discussion to include a comparison of the findings with previous studies testing PERK inhibitors or chemical chaperones in other neurodegenerative models. Furthermore, addressing the lack of efficacy observed for the tested compounds and suggesting alternative approaches (e.g., targeting different pathways or exploring combination therapies) would be valuable.

The pharmacokinetic findings could also benefit from further exploration. While the manuscript touches on the concentrations of TUDCA used in the CHOP::luciferase assays, a deeper explanation of how these concentrations compare to those used in vivo would help clarify whether the lack of effect observed in vivo could be due to non-physiological concentrations in vitro.

Comment #5:

The authors mention plans to explore trazodone’s mechanism of action further. I recommend expanding on this point by suggesting possible future studies that could overcome the limitations identified in this work. This might include exploring alternative models or approaches to better understand the drug’s effects in the context of MSS.

Comment #6:

The manuscript would benefit from a thorough review to address typographical and grammatical errors. Ensuring consistency in grammatical structure, particularly in the use of past and present tenses, and maintaining varied yet appropriate terminology would enhance readability. Avoiding repetitive use of the same words, such as "we," throughout the text would further improve the overall flow and clarity.

6. PLOS authors have the option to publish the peer review history of their article (what does this mean?). If published, this will include your full peer review and any attached files.

Reviewer #1: No

Reviewer #2: **Yes: **Muhammad Ateeb

---

## [Author Response · Author response to Decision Letter 0]

18 Dec 2024

Reviewer #2

Specific Comments:

Comment #1:

The abstract is detailed but could be made clearer or more focused. I suggest emphasizing the current primary findings, specifically the lack of neuroprotective effects, and discussing their potential implications for future MSS research if possible. This would provide readers with a clearer and more focused overview of the study’s outcomes.

We have revised the abstract to more clearly emphasize the lack of neuroprotective effect on Purkinje cell degeneration. We have also included a new sentence discussing the implication of our findings for future MSS research.

Comment #2:

I suggest the author to provide more context for selecting trazodone, DBM, and TUDCA as the focus of this study. Expanding on the rationale behind testing these compounds specifically in MSS, as well as discussing the role of the PERK pathway in MSS pathophysiology, would strengthen the manuscript. It would also help to explain why these drugs were hypothesized to be effective in this context.

We have clarified the rationale for selecting trazodone, DBM, and TUDCA in the Introduction, referencing our perspective article published in Neural Regeneration Research (ref. 12), where we explain why these drugs were hypothesized to be effective in the context of MSS. Further details about the rationale for using trazodone and DBM are provided in the second paragraph of the Discussion, citing previous studies demonstrating their partial PERK inhibitory activity and neuroprotective effects in other neurodegenerative disease mouse models. We believe the current manuscript provides sufficient context for the study's objectives and would prefer not to add additional text to avoid redundancy or verbosity, unless the Editor advises otherwise.

Comment #3:

While the manuscript includes a power analysis, it would benefit from a more detailed justification for selecting the 60% reduction in missteps as the target effect size. Clarifying this decision would improve the transparency and credibility of the study’s design. Additionally, further discussion on the rationale for the chosen dosing regimens particularly in relation to previous studies or pharmacokinetic data would provide a clearer foundation for the experimental approach.

The manuscript could also clarify the relevance of the CHOP::luciferase assay to the in vivo findings, especially for readers who may be less familiar with molecular techniques. Finally, I encourage the authors to emphasize the translational challenges identified in the study and offer recommendations for future drug testing in MSS models.

We set a target effect size of a 60% reduction in missteps, as this approximates the effect previously observed when woozy mice were treated with the PERK inhibitor GSK2606414. This rationale has now been explicitly stated in the Study Design section of the Materials and Methods.

Regarding the dosing regimens, these were chosen based on previously published studies in neurodegenerative disease models, as discussed in the manuscript. We believe this provides sufficient justification for the selected regimens.

The CHOP::luciferase assay is a well-established method for assessing the effects of compounds on PERK pathway activation, and was included to validate the compounds’ activity at the molecular level. However, translating these results to the in vivo setting in terms of predictive efficacy and drug dosing is challenging, as we now state in the revised Discussion.

We have emphasized the translational challenges identified in the study and offered alternative approaches (see points 1 and 4).

Comment #4:

I recommend expanding the discussion to include a comparison of the findings with previous studies testing PERK inhibitors or chemical chaperones in other neurodegenerative models. Furthermore, addressing the lack of efficacy observed for the tested compounds and suggesting alternative approaches (e.g., targeting different pathways or exploring combination therapies) would be valuable.

The pharmacokinetic findings could also benefit from further exploration. While the manuscript touches on the concentrations of TUDCA used in the CHOP::luciferase assays, a deeper explanation of how these concentrations compare to those used in vivo would help clarify whether the lack of effect observed in vivo could be due to non-physiological concentrations in vitro.

We believe the current manuscript provides a focused discussion of previous studies using PERK inhibitors and the chemical chaperone TUDCA in neurodegenerative disease mouse models. To maintain this focus, we prefer not to further expand the discussion to include additional chemical chaperones, as suggested by the reviewer.

The manuscript already discusses alternative approaches, such as the use of a small-molecule Akt activator to enhance PERK phosphorylation at threonine 799 and the potential of gene therapy to restore SIL1 function. Given the limited understanding of other pathogenic mechanisms in MSS beyond maladaptive PERK-mediated UPR activation, discussing additional pharmacological strategies or combination therapies would be speculative and beyond the scope of this study.

We agree with the reviewer that the >250 µM concentration of TUDCA active in the CHOP::luciferase assay is substantially higher than the submicromolar concentrations achieved in the brain of mice treated with the dosing regimens used in our study (0.4% in the diet or 500 mg/kg intraperitoneally). These dosing regimens were selected based on their demonstrated efficacy in previous neurodegenerative disease mouse models (refs 20, 21, 24-30). As explained in our response to Comment #3, the results obtained in vivo might not be directly related to the in vitro results in the CHOP::luciferase assay.

Comment #5:

The authors mention plans to explore trazodone’s mechanism of action further. I recommend expanding on this point by suggesting possible future studies that could overcome the limitations identified in this work. This might include exploring alternative models or approaches to better understand the drug’s effects in the context of MSS.

In the manuscript, we speculate that the slight improvement in beam walking performance observed in trazodone-treated woozy mice may be related to its potential effects on serotonergic cerebellar transmission. To address the reviewer’s suggestion, we have added a brief sentence outlining how this hypothesis could be tested experimentally in future studies.

Comment #6:

The manuscript would benefit from a thorough review to address typographical and grammatical errors. Ensuring consistency in grammatical structure, particularly in the use of past and present tenses, and maintaining varied yet appropriate terminology would enhance readability. Avoiding repetitive use of the same words, such as "we," throughout the text would further improve the overall flow and clarity.

The manuscript has been reviewed by a UK English editor with extensive experience in editing scientific papers.

---

## [Editor Report · Decision Letter 1]

29 Dec 2024

Trazodone, dibenzoylmethane and tauroursodeoxycholic acid do not prevent motor dysfunction and neurodegeneration in Marinesco-Sjögren syndrome mice

PONE-D-24-30565R1

Dear Dr. Roberto Chiesa,

We’re pleased to inform you that your manuscript has been judged scientifically suitable for publication and will be formally accepted for publication once it meets all outstanding technical requirements.

Kind regards,

Shahbaz Ahmad Zakki

Academic Editor

PLOS ONE
---

## [Editor Report · Acceptance letter]

3 Jan 2025

PONE-D-24-30565R1 

PLOS ONE

Dear Dr. Chiesa, 

I'm pleased to inform you that your manuscript has been deemed suitable for publication in PLOS ONE. Congratulations! Your manuscript is now being handed over to our production team.

Kind regards, 

on behalf of

Dr. Shahbaz Ahmad Zakki 

Academic Editor

PLOS ONE